# Research on Micro-Grid Group Intelligent Decision Mechanism under the Mode of Block-Chain and Multi-Agent Fusion

**Xiaolin Fu [1], Hong Wang [2], Zhijie Wang [1,\*], Zhong Shi [3]**  **, Wanhao Yang [1] and Pengchi Ma [1]**

[1] College of Electrical Engineering, Shanghai Dianji University, Shanghai 201306, China; 186001010319@st.sdju.edu.cn (X.F.); 186001010415@st.sdju.edu.cn (W.Y.); 186001010408@st.sdju.edu.cn (P.M.)

[2] School of Economics & Management, Tongji University, Shanghai 200092, China; wanghongpy@163.com

[3] New Bei-yang Information Technology Co., Ltd., Weihai 264200, China; shizhongsd@126.com

\* Correspondence: wangzj@sdju.edu.cn; Tel.: +86-189-6458-6826

**Abstract:** This paper aims to study the problems of surplus interaction, poor real-time performance, and excessive processing of information in the micro-grid scheduling and decision-making process. Firstly, the micro-grid dual-loop mobile topology structure is designed by using the method of block-chain and multi-agent fusion, realizing the real-time update of the decision-making body. Secondly, on the basis of optimizing the decision-making body, a two-layer model of intelligent decision-making under the decentralized mechanism is established. Aiming at the upper model, based on the theory of block-chain consensus mechanism, this paper proposes an improved evolutionary game algorithm. The maximum risk-benefit in the decision-making process is the objective function, which realizes the evaluation and optimization of decision tasks. For the lower layer model, based on the block-chain distributed ledger theory, this paper proposes an improved hybrid game reinforcement learning algorithm, with the maximum controllable load participation as the objective function, and realizes the optimal configuration of distributed energy in the micro-grid. This paper reveals the rules of group intelligent decision making in micro-grid under multi-task. Finally, the effectiveness of the proposed algorithm is verified by using Beijing Jin-feng Energy Internet Park data.

**Keywords:** block-chain; multi-agent; evolutionary game algorithm; group intelligence decision

## 1. Introduction

In 2015, the European Union-funded Intelligent Grid Integrated Research Project ELECTRA IRP (European Liaison on Electricity Committed Towards long-term Research Activity Integrated Research Programme) proposed the concept of "Web-of-Cells (WoC)". Since then, people have discussed how to implement the overall optimal scheduling and control strategy for a class of complex systems based on a large number of group units with limited information, weak controllability, small capacity, and wide distribution [1]. This paper focuses on the direction of multi-cooperative evolutionary game between distributed energy sources and group intelligent decision-making theory in micro-grid. Aiming at this problem, the current common solution is to use a multi-agent system (MAS) with good autonomy to solve the problem of collaborative optimization of distributed systems by using its independent and parallel computing features.

At present, by consulting the existing literature, I found that the MAS can optimize the operating characteristics of the micro-grid mainly in the following 10 points: (a) Under the constraints of input saturation and communication time delay, the control protocol is designed and the appropriate control input parameters are selected to achieve the leader's follow-up consensus control, which effectively

reduces the number of updated controllers and improves the stability of system operation [2]. However, the number of controllers that are reduced is not time-varying, and adjustments cannot be made in time according to different operating conditions in each period, so the system cannot always be kept in an optimal state. (b) Design an agent organization that includes both centralized and distributed concepts. When there is a delivery line, the number of deferred tasks is greatly reduced, and time delay can be effectively avoided [3]. However, this solution only solves the problem of time delay in a specific scenario and is not universal. (c) By constructing the controller reasonably, the cooperative tracking of all agents is realized, and the event trigger control mechanism is introduced to reduce the communication burden and improve the system resource utilization [4]. However, there is a problem of interactive redundancy in the process of collaborative tracking. Each follower must interact with all followers under the same leader, increasing system energy consumption and reducing the communication efficiency of multi-agent when making decisions. (d) By using the specified time optimal control, a new containment tracking algorithm is proposed to provide time-varying feedback gain for MAS. Resolve the containment tracking problem that occurs within a predetermined settling time by designing a sampling time series that converges to a specified time [5], but the proposed algorithm is based on the specific time containment tracking algorithm of the sampled data. The amount of sampled data that needs to be processed in the early stage is large, which is not conducive to improving the execution efficiency of the algorithm. (e) By using the relative coupling control structure to improve the cooperative performance of the MAS, the network predictive control scheme is adopted to actively deal with network delay and data loss, and the flexibility of the MAS is improved [6]. However, the stability of the networked collaborative MAS has not been studied, and the engineering significance of the proposed strategy cannot be effectively verified. (f) At present, for the consistency problem caused by the mutual position difference between single agents in the cluster process, a cluster control algorithm driven by local position observation and adaptive weighting coefficient is proposed to reduce the influence of position difference on agent speed evaluation and improve the MAS output characteristics [7]. However, the impact of time delays on the clustering process of distributed decision makers is not considered. (g) The use of a consistent algorithm to achieve optimal power allocation can improve the robustness of the system [8,9]. (h) The use of adaptive optimal control and distributed output feedback can reduce the communication complexity of MAS and improve the optimization efficiency. [10–12]. (i) Improve the stability of MASs by eliminating dead zones and solving consensus functions [13,14]. (j) Synergistic control between agents is improved by dynamically correcting the consensus mechanism [15–18].

The interaction redundancy between multiple agents is still the key factor that restricts the convergence of the system. At the same time, the communication delay and the amount of information processed greatly affect the speed at which the system converges consistently, which will reduce the ability of the MAS to perform multi-tasking. This paper proposes a micro-grid group intelligent decision-making mechanism combining block-chain and multi-agent, using peer-to-peer network decentralized by block-chain, consensus mechanism for rapid response to distributed decision-making, and smart contract with high security and high efficiency. Design micro-grid double-loop mobile topology to optimize decision-making body, the distributed intelligent decision-making two-layer model is built to make decision task evaluation and decision task assignment, and improve the communication efficiency of the MAS during power system supply and demand interaction.

The main innovations of this paper can be summarized as follows:

1. Improve the traditional way of topology optimization using distributed algebraic algorithm. Innovatively, the intelligent contract is guided by the optimal adjustment of the controllable load, and the micro-grid dual-loop mobile topology model is built by the combination of block-chain and multi-agent. It improves the complexity of communication topology and reduces the redundant interaction between agents.

2. Design a fast selection strategy of power supply nodes based on consensus mechanism, and use the double-loop matching degree function to correct the decision body determined by the

dual-loop mobile topology model of the micro-grid in real time. It is in line with the actual running condition of the node and has guiding significance for the actual engineering planning problem.

3.  Set up a bi-level model of group intelligent decision-making under the decentralized mechanism. The upper layer takes the maximum risk-reward amount as the optimization goal, and on this basis, innovatively introduces the distributed account book and consensus mechanism to improve the evolutionary game algorithm in the block-chain. Under the premise of ensuring the credibility of individual decision-making, the task rationality is evaluated; the lower level is controlled by the maximum controllable load, and the hybrid game is used to solve the learning algorithm. By introducing the block-chain decentralized block decision, non-convex optimization problem for multi-task decision making with high quality.

4.  Using the above model to optimize the intelligent decision-making of controllable load groups in Beijing Jin-feng Micro-grid Park.

## 2. Multi-Agent Interaction Network Model

### 2.1. Interaction Model

The microgrid system includes a distributed generator, energy conversion device, energy storage device, controllable load, monitoring device, etc. It is a small energy management system [19]. As an important part of smart grid energy management, smart power participates in demand response through real-time interaction with power flow, information flow and business flow on the grid side [20], among which MAS. It is widely used in demand response because it enables micro-sources to rely on local controllers for autonomous and coordinated control [21]. As shown in Figure 1, each component in the actual microgrid can be equivalent to an intelligent body, and the connection state with the microgrid is judged by the switch on and off. For example, ES represents an energy storage agent, DG represents a distributed energy agent, Opt. represents an economic dispatch optimization agent, and Load represents a load agent. This paper mainly studies the regulation of load intelligence in the microgrid under the multi-agent structure. By adopting a distributed control structure, communication with adjacent units is realized, the dependence on the communication system is reduced, the system's robustness is enhanced, and the user is encouraged to dynamically adjust the power consumption according to the balance of power supply and demand.

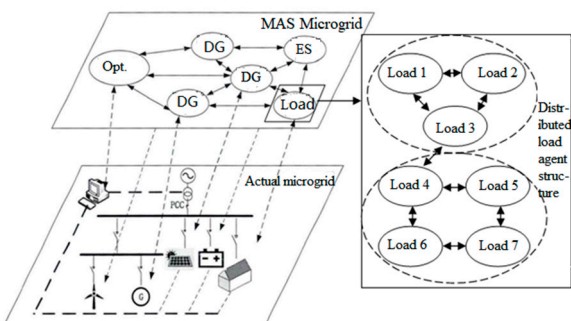

**Figure 1.** Microgrid and MAS corresponding structure diagram.

In the MAS mode, in order to maintain the stability of the distribution network voltage, the upstream N-1 node transmits the voltage cost information to the downstream, combined with the operating conditions at the node N, generates power cost information and returns to the end of the feeder node, passing the voltage cost information and the power cost. The propagation of information within the MAS changes the working state of the flexible load at each node in the distribution network to meet the system objectives. The environment variable set $E$ contains a plurality of environment variables $e$, as shown in Equation (1).

$$E = \{e, e', \ldots\} \tag{1}$$

The interaction process between the agent and the external environment is realized by executing an action change environment variable, and the action set $A_c$ includes a plurality of actions $\alpha$ that can change the variables of the external environment, as shown in the Formula (2). The action $\alpha$ affects the environment variable $e$. The new environment variable as an output affects the next action of the agent. This cycle continues until the agent completes the target task, that is, the external environment variable reaches the ideal state set by man [22].

$$A_c = \{\alpha, \alpha', \ldots\} \tag{2}$$

*2.2. Flexible Load Model*

The reason why the demand response can improve the grid's ability to accept large-scale renewable energy is that the demand response changes the time matching between the load and the renewable energy generation, and realizes the two-way interaction between the grid and the user [23]. Therefore, starting from the load response characteristics, this paper firstly divides the internal load of the microgrid into four categories: translatable load, transferable load, load reduction, and interruptible load, and considering the decision under the distributed MAS structure, it can be translated. The adjustment characteristics of the load-to-trigger time are combined with the charge–discharge characteristics of the temperature-controlled energy storage to reduce the load, and the typical electrical equipment in the two types of loads is further integrated into an overall scheduling unit, and the decision is made therein. Under the condition of consistency, each unit can minimize the power consumption during peak hours/high electricity prices, optimize the load curve and decision mechanism. This article selects the translational load: washing machine, dishwasher, microwave oven, and load reduction: air conditioner, electric water heater, the combination is as follows:

1.  {Washing machine + Air conditioning} Agent
2.  {Dishwasher + Electric water heater} Agent
3.  {Washing machine + Dishwasher + Air conditioning} Agent
4.  {Washing machine + Air conditioner + Microwave + Electric water heater} Agent

After the grid company announces the required response capacity, each agent performs a demand response adjustment metric reporting strategy based on its online energy consumption, assuming that the discrete system dynamic response of each overall scheduling unit is [24]

$$x_i(k+1) = x_i(k) + u_i(k) \tag{3}$$

where $x_i(k)$ is the local information of the agent $i$ when iterating $k$ times; $x_i(k+1)$ is the local information of the agent $i$ when iterating $k+1$ times; $u_i(k)$ is the input of the distributed control.

Under the MAS distributed architecture, the local communication protocol with shiftable load and load reduction can be expressed as

$$u_i(k) = \frac{1}{1+d_i} \sum_{j \in N_i} a_{ij}(x_j(k) - x_i(k)) \tag{4}$$

where $d_i$ is the node weight of node $i$. Whether a node can obtain decision-making power depends on the node weight. In the decision-making process, the price-required response can be reduced by reducing the load, that is, the user adjusts the demand for electricity according to the change of the retail price, actively changes the original power consumption mode, and realizes the expected adjustment function by means of demand elasticity [25], through local agreement and the load can be translated to communicate, and the electricity price elasticity $e_{st}$ of the power is defined as

$$e_{st} = \frac{\Delta L_s / L_s^0}{\Delta P_t / P_t^0}, e_{st} \le 0, s = t; e_{st} > 0, s \ne t \tag{5}$$

where $e_{st}$ is the price elasticity of time $s$ to time $t$; $L_s^0$ and $P_t^0$ are the electricity load at the time $s$ before the price demand response and the electricity price at time $t$, respectively, as the execution basis of the decision; $\Delta L_s$ and $\Delta P_t$ are the time after the demand response The amount of load change of $s$ and the amount of price change of time $t$. The load change of the user participating in the price type demand response is

$$
\begin{bmatrix}
\Delta L_1 / L_1^0 \\
\Delta L_2 / L_2^0 \\
\vdots \\
\Delta L_{24} / L_{24}^0
\end{bmatrix}
=
\begin{bmatrix}
e_{1,1} & \cdots & e_{1,24} \\
\vdots & & \vdots \\
e_{24,1} & \cdots & e_{24,24}
\end{bmatrix}
\begin{bmatrix}
\Delta P_1 / P_1^0 \\
\Delta P_2 / P_2^0 \\
\vdots \\
\Delta P_{24} / P_{24}^0
\end{bmatrix}
\tag{6}
$$

The power value of the user's consumed electricity $L_t$ is defined as $V(L_t)$, $L_t^0$ is the load demand at the time $t$ before the demand response, and the load model under the distributed agent structure is

$$
L_t = L_t^0 \{ 1 + e_{tt} \frac{L_t - L_t^0}{L_t^0} + \sum_{t=1,s \neq t}^{24} e_{st} \frac{P_t - P_t^0}{P_t^0} \}
\tag{7}
$$

## 3. Double-Loop Mobile Topology in Block-Chain and Multi-Agent Fusion Mode

Distributed topology optimization algorithm based on minimum weighted rigid graph can theoretically solve the problem of difficult graphics operation. However, due to the increasing non-linearity of MAS, uncertainties in parameters and time delays, the information interaction between MAS is affected [26]. Therefore, the choice of decision-makers is particularly important. This paper proposes a dual-loop mobile topology of micro-grid in the mode of block-chain and multi-agent fusion, and builds a decentralized decision topology model. The block-chain can quickly respond to the characteristics of distributed decision-making, and the fast selection of power supply nodes can be realized through the consensus mechanism, and use the double-loop matching function to perform real-time compensation correction, to meet the actual operation requirements to the greatest extent, improve the communication topology complexity and reduce the redundant interaction between agents.

### 3.1. Topology Execution Rules

Resilient packet rings (RPR) can inspire the bandwidth utilization and effective bandwidth by dynamically allocating bandwidth, so that devices on the ring share all or part of the bandwidth. This paper combines block-chain and multi-agent design micro-grid. The dual-loop mobile topology works as follows.

The inner ring is the basic ring, which comprehensively considers the complementary characteristics of the translatable load and the temperature-controlled energy storage load, and the part of the decentralized trusted transaction environment that can be established by the alliance chain in the block-chain, which can convert translatable load and load reduction fixed reorganization as an overall decision unit, as shown in Figure 2 and Table 1. Assuming that the power imbalance in the micro-grid is generated by Distributed Generator 1 (DG1), it can be adjusted through three private chains of BC3, BC4, BC5, and a joint chain of BC6, and the decision weight is 0. The unit can be used without force. During the reorganization, the load controllability and switching speed are improved to reduce the electricity consumption during peak hours/high electricity prices and to meet the demand for electricity in different time periods. The outer ring is a correction ring, and the decision body is corrected by guiding the real-time controllable load adjustment amount.

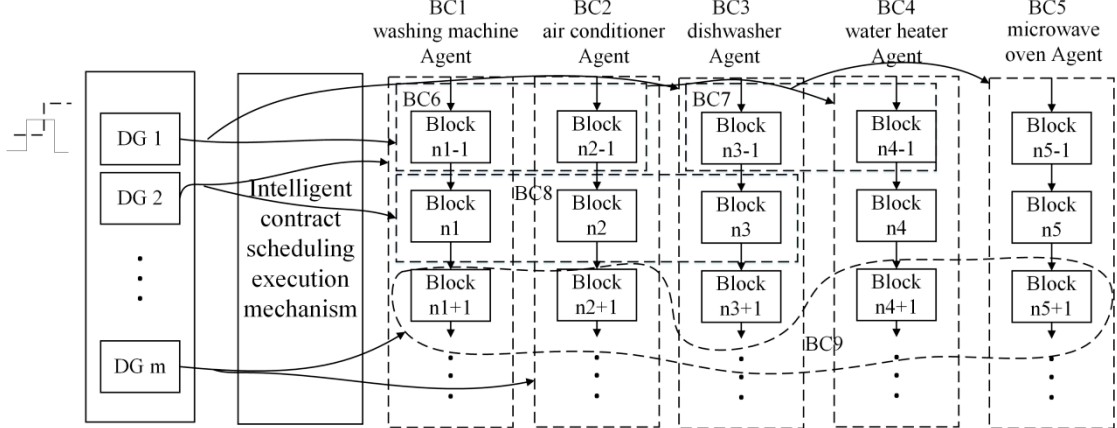

**Figure 2.** Inner loop mobile topology.

**Table 1.** Inner ring mobile topology.

| Private Chain | Alliance Chain |
|---|---|
| Washing machine decision chain (BC1) | Washing machine + Air conditioning transaction chain (BC6) |
| Air conditioning decision chain (BC2) | |
| Dishwasher decision chain (BC3) | Electric water heater + dishwasher trading chain (BC7) |
| Electric water heater decision chain (BC4) | Air conditioner + Washing machine + Dishwasher trading Chain (BC8) |
| Microwave oven decision chain (BC5) | Washing machine + Microwave + Air conditioning + Electric water heater trading chain (BC9) |

In this paper, we use the topology discovery algorithm and the distributed differential evolution algorithm DE/rand/1/bin to write intelligent contracts. The decision-making subject selection model is shown in Equations (1) to (4). The topology discovery algorithm enables each agent to obtain the necessary network topology information. The distributed differential evolution algorithm is a heuristic search algorithm based on population evolution. The DE/rand/1/bin difference strategy has strong global search ability and is easy to maintain the population. Diversity is easy to obtain the characteristics of the optimal solution.

1.  Agent *i* Receive data through neighbors

$$r_i[l+1] = U_{i \in N_i}\left\{T_i[k] \cap s_{i,j}[l]\right\} \tag{8}$$

where *r* denotes association information received by the agent in the topology; *i* and *j* are the decision bodies separated by the inner ring, respectively, and as a multi-agent, *l* is the number of iterations of the topology discovery algorithm, and *k* is the number of iterations of the distributed differential evolution algorithm DE/rand/1/bin, $T_i[k]$ is the communication topology of the system at the *k*th iteration of the agent *i*, and $s_{i,j}[l]$ is the load adjustment amount obtained by the agent *i* from the neighbor agent *j*.

2.  Mismatch estimation

$$p_i[n+1] = p_i[n] + \varepsilon \sum_{j \in N_i} a_{i,j}(p_j[n] - p_i[n]) \tag{9}$$

where $p_i$ is the local estimate of the power mismatch, $a_{i,j}$ is the average power mismatch of the agent *i*, *j*, and $\varepsilon$ is the branch step size. When the $\varepsilon > 1/\max_i l_{ii}$ is satisfied, the decision body in

the agent $i$ needs to be corrected by the outer ring, where $l_{ii}$ is the current flowing through the agent, and $\varepsilon$ is used as an indicator for judging whether to perform the outer ring correction.

3.　Update the local action value

$$\widetilde{J}(S_{N_i,t}, P_{N_i,t}) \leftarrow (1-\omega)\widetilde{J}(S_{N_i,t}, P_{N_i,t}) + \omega[c_{i,t} + \gamma \min_{P_{N_i,t}} \widetilde{J}(S_{N_i,t+1}, P_{N_i,t})] \tag{10}$$

where $S_{N_i,t}$ is the joint state of the neighbor agent, $P_{N_i,t}$ is the joint action of the neighbor agent, $\omega$ is the fitness when the action $P_{N_i,t}$ is taken at the $k'$th time in state $S_{N_i,t}$, $\omega = \omega_{k'}(S_{N_i,t}, P_{N_i,t})$, $c_{i,t}$ is the average power generation cost estimate, and $\gamma$ is the agent's output ratio, $\widetilde{J}$ is the updated local power value.

4.　Feasible power generation combination

$$P'_{i,t} = P_{i,t} + \sum_{j \in N_i} a_{i,j}(\delta_j - \delta_i) \tag{11}$$

where $\delta_i$ and $\delta_j$ are the controllable output power values of the agent $i$ and the agent $j$, respectively, and the value is between $[-\overline{P}_i, \overline{P}_i]$.

When the transmission power loss between the agents is negligible, the power generation combination given by Equation (4) can satisfy the power balance constraint because $\sum_{i \in I} \sum_{j \in N_i} a_{i,j}(\delta_j - \delta_i) = 0$.

The decision process is shown in Figure 3. When there is a power imbalance due to the deviation of the distributed energy power prediction, the fixed decision unit that has been planned by the inner loop is preferentially used for adjustment. If a new balance of power can be achieved, the adjustment is ended, determining the optimal output combination at the current moment; and if the power is still unbalanced, the consensus mechanism is used to realize the fast selection of the node and the matching degree is used to calculate the matching degree of the two rings. The power output value on the contact line is used to correct it to determine whether the current time output demand is met. If it is satisfied, it is output as the current optimal choice. If it is not satisfied, the consensus mechanism is used to quickly select the node and correct the matching function and iterate until the power is balanced in the micro-grid.

As shown in Figure 4, the air conditioner is represented by the number 10 in the schematic diagram, the electric water heater is represented by 11, the washing machine is represented by 01, the dishwasher is represented by 12, the microwave oven is represented by 00, and the power balance is achieved by default inside the same decision body connected at the same point. Therefore, the decision-making body in the figure is BC6, BC7, BC8, BC9 in Table 2 in counterclockwise order. When power imbalance occurs, the controllable load in the inner loop is preferentially used for power adjustment. If there is still power shortage, the load can be adjusted as a guide, and the node is quickly selected and matched by the consensus mechanism. The function is modified to treat the inner and outer loop decision units as a whole for collaborative decision making. The outer ring dotted box represents the selectable decision subunits, which are added or deleted as the inner loop real-time power changes.

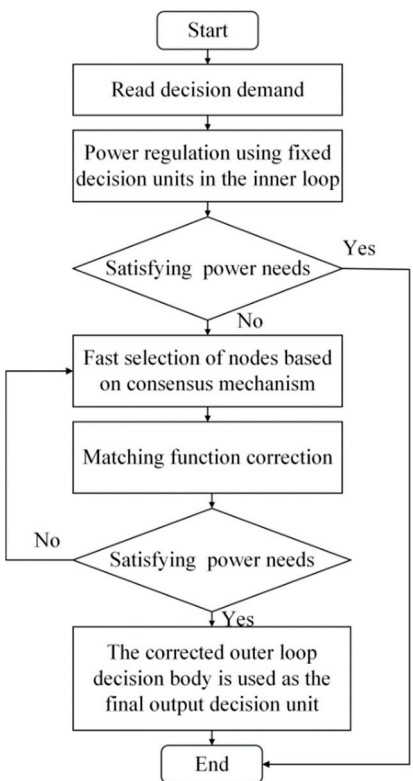

**Figure 3.** Decision flow chart.

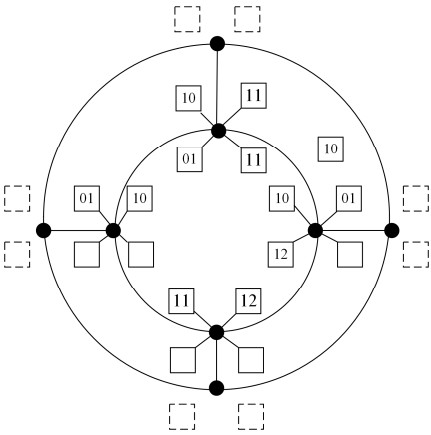

**Figure 4.** Schematic diagram of double-ring topology.

*3.2. Fast Selection Strategy for Nodes Based on Consensus Mechanism*

3.2.1. Consensus Mechanism Competition Billing Rights

The block-chain solves the problem of transmitting trusted information and value transfer on untrusted channels. As the core technology of the block-chain, the consensus mechanism can safely update the data state in the distributed network and reach agreement in a distributed scenario. Realizing the rapid recording and verification of information provides good support for solving the trust problem and fast response problem among distributed nodes in the proposed double-ring structure. The implementation steps of the consensus mechanism are as follows:

1. The transfer transaction is sent to the Bit-coin network, and all the mining pools collect all transaction records;
2. Each pool calculates the random number *nonce* on the basis of the original transaction data and serves as the block header;
3. Constantly adjusting the *nonce* value so that the entire block calculates the *Hash* value that satisfies the consensus rule;
4. Once a certain mine pool calculates the correct *Hash* value, the mining is successful. The mine pool obtains the billing rights and corresponding rewards. It broadcasts through the whole network and informs other mine pools to confirm that it has been added to its own block list.
5. Receive the next transfer record.

### 3.2.2. Node Quick Selection Strategy

1. Judging that the decision-making body of the outer ring can adjust the output change value.

$$|AOP - DOP|_t \geq \xi \tag{12}$$

where *AOP* is the output power of each decision-making body of the outer ring, which changes continuously with time. *DOP* is the demand output power in the micro-grid, which changes continuously with time. $\xi$ is the control parameter, the analog function is the same as the Hash value in the consensus mechanism, and can play a guiding role, $\xi = [(1 - (\frac{1}{2})^n) \cdot a_{i,j}]$.

2. Calculate the probability of stability of the adjustable output

$$b_i = S_i^{t+1} / \sum_{i \in RT_i^{t+1}} S_i^{t+1} \tag{13}$$

where $S_i^{t+1}$ is the stable output power value of the outer ring agent *i* at time $t+1$.

3. Estimate the entropy value and determine the node

$$H_i(\Delta_i) = -\sum b_i \log b_i / \log N_i \tag{14}$$

where $N_i$ is the number of neighbor nodes of the agent $i, N_i = \left| RT_i^{t+1} \right|$.

The higher the entropy value, the more stable the node. When selecting the next decision node, the node first calculates its own entropy according to Equation (14) and then broadcasts it to the neighbor node. Once the node receives this information, it saves this information in the multi-agent block and then selects the most stable node as the next decision node.

### 3.3. Matching Degree Function Correction

$$MT_i = [\chi_i \cdot \gamma + (1 - \chi_i) \cdot \gamma'] / a_i \tag{15}$$

where $\chi_i$ is the number of adjustments of the agent *i* in the inner ring. $\gamma$ is the output ratio of the agent. $\gamma'$ is the ratio of the corrected agent output. $a_i$ is the power mismatch of the agent *i*. $MT_i$ is the matching degree determined by the agent *i* after the function correction, and the decision body of the next round is determined according to the matching degree.

### 3.4. Validation Check

In order to verify the effectiveness of the dual-ring mobile topology described above, the distributed power supply in Beijing Jinfeng Energy Internet Park is taken as an example to design a dual-loop design for its internal distributed energy. The distributed energy parameters are shown in Table 2:

**Table 2.** Beijing Jin-feng Energy Internet Park equipment parameters.

| Object | Parameter Type | Value |
|---|---|---|
| Wind Turbines | Cut-in wind speed (m/s) | 4 |
| | Cut-out wind speed (m/s) | 25 |
| | Rated wind speed (m/s) | 12 |
| | Power Coefficient | 0.60 |
| Photovoltaic Array | Power-temperature coefficient | 0.043% |
| | PV array panel temperature (°C) under standard conditions | 45.5 |
| | Number of series units of PV modules | 70 |
| | Number of parallel units of PV modules | 30 |
| Diesel generators | Rated power (kW) | 200 |
| | Minimum output (kW) | 16 |
| Energy storage battery | Rated capacity (kWh) | 500 |
| | SOC min | 0.2 |
| | SOC max | 0.9 |
| | Charge/discharge power (kW) | −60/60 |
| | Charge/discharge efficiency | 0.92/0.93 |
| Distribution network | Interactive power (kW) | −300/400 |

The abbreviations in Figures 5–8 represent distributed power sources, for example: PV for photovoltaic arrays, ES for energy storage, WT for wind turbines, and DE for diesel generators. The designed inner ring topology includes: 'WT+ES' autonomous chain, 'PV+ES' autonomous chain, 'PV+WT+ES' trading chain, and 'PV+WT+ES+DE' trading chain. It can be corrected by the outer ring under the guidance of adjustable output, which plays an active role in the supply and demand interaction of energy in the micro grid. The following figure shows the dynamic decision subjects identified by four typical moments:

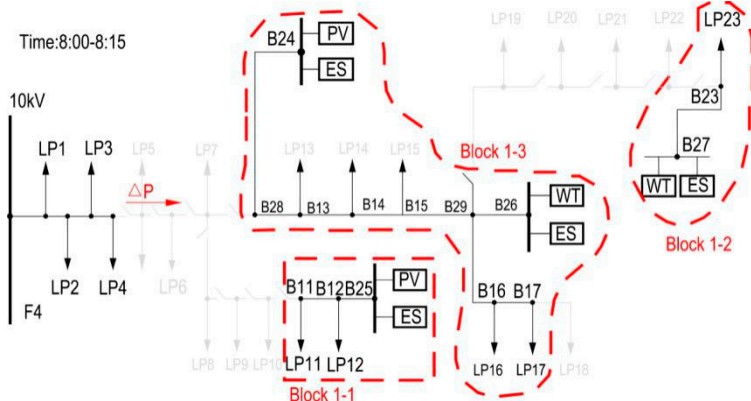

**Figure 5.** 8:00–8:15 division of decision-making body.

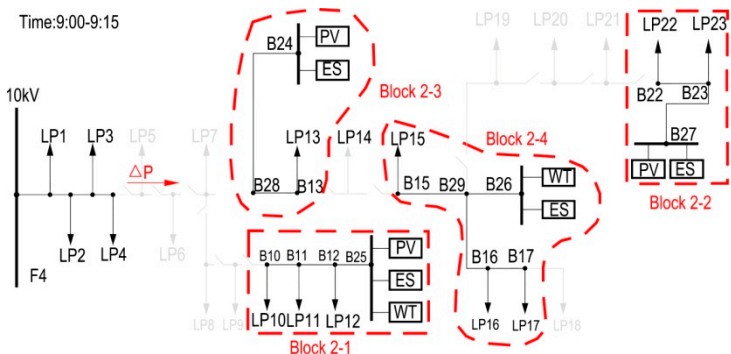

**Figure 6.** 9:00–9:15 division of decision-making body.

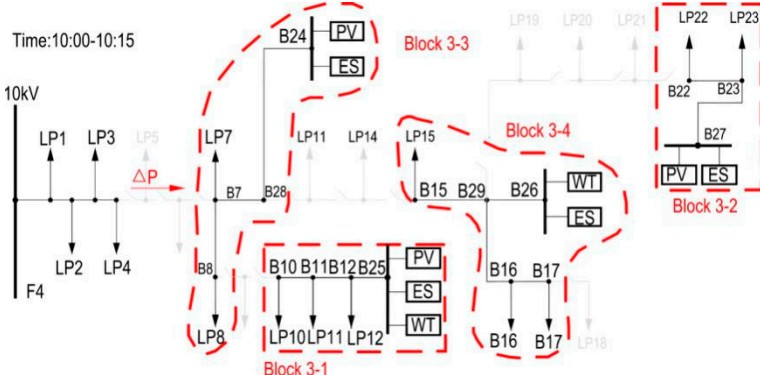

**Figure 7.** 10:00–10:15 division of decision-making body.

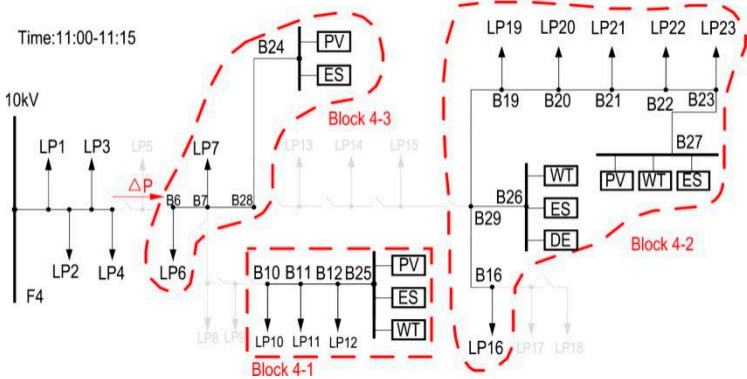

**Figure 8.** 11:00–11:15 division of decision-making body.

It can be seen from Figure 5 to Figure 8 that the topology detection algorithm and the distributed differential evolution algorithm DE/rand/1/bin are used to write the inner loop decision body determined by the smart contract, and in most cases, the power imbalance in the micro grid can be adjusted. In Figure 8, the inner loop decision body still has a power shortage of 3.5 kW after power adjustment, and the power demand is not met. Therefore, the outer ring is introduced for correction. Among them, the micro gas turbine located 300 m away from the Block 4-1 and the diesel generator 100 m away from the 'wind storage' system in Block 4-2 can still be adjusted for power. The stability probability of the two is calculated as: 82.5% and 76.3%, entropy the values are 0.693 and 0.802. The diesel generator node with high entropy and good stability probability is selected to modify the inner loop decision-making unit. The new decision-making body is 'wind turbines + energy storage + diesel generator', which can meet the power demand after correction.

All of the models and control strategies described in this paper are under the 'MAS+ block-chain' structure, which has no significant improvement in robustness and reliability for a single agent

compared to a single agent, it can significantly improve the self-learning ability and communication efficiency of the agent, and achieve accurate calculation of the ratio of decision-making output and efficient configuration of distributed energy.

It can be seen from Table 3 that the multi-agent and block-chain fusion mode uses the topology discovery algorithm and the distributed differential evolution algorithm DE/rand/1/bin to write a smart contract to determine the process of the inner loop decision subject, using the complementary characteristics of the load. The establishment of the decentralized trusted transaction environment, fully playing the role of the distributed power supply in the micro-grid and reducing the complexity of the network topology, taking 11:00–11:15 decision-making body as an example, can reduce communication 5 times. The channel redundancy is reduced by 36%, which ensures the rationality of the decision-making body. The outer loop is guided by the optimal power adjustment of the controllable load. The consensus mechanism can quickly respond to the characteristics of distributed decision-making and realize the rapid selection of the power supply node. Improve communication efficiency by 8.6%, reduce redundant interaction between agents, and improve the ability of decision-making subjects in group intelligent decision-making.

**Table 3.** Comparison of indicators after introducing dual-ring mobile topology.

| Decision Subject | Number of Interactions | Redundancy Rate | Communication Efficiency |
|---|---|---|---|
| Multi-agent | 11 | 1.33 | 68.1% |
| Multi-agent+ block-chain | 4 | 0.97 | 76.7% |

## 4. Group Decision-Making Bi-Level Model under the Decentralized Mechanism

Intelligent decision-making is the combination of decision support system (DSS) and artificial intelligence technology. The idea of artificial intelligence knowledge representation and knowledge processing is introduced into DSS [27], which can effectively overcome the traditional decision-making learning ability, data processing speed, and system decision-making. Insufficient efficiency, with the deepening of smart grid construction and the increase of distributed new energy penetration rate on the user side, it is of great practical significance to study the intelligent decision model with fast response and high reliability. The group intelligent decision-making can promote the interaction between the grid and the user energy flow, information flow, and business flow on the basis of optimizing the micro-grid operation structure [28] and improving the rationality of resource allocation [29]. This paper establishes an upper-level decision-making model with the maximum risk-reward goal. In the upper-level decision-making planning, the impact of individual decision-making credibility on the operational strategy is considered, and the consensus mechanism is introduced and transformed into the upper-level constraint. The maximum controllable load participation is taken as the lower target, and two ways to adjust the load participation degree are: (1) price incentive; (2) demand side bidding converts the influence of the optimal operation mode under different scenarios of the micro grid into the lower layer constraint. The decision objects in the two-layer model refer to the subjects based on the double-loop mobile topology.

### 4.1. Upper Decision Model

In the current smart grid decision-making, the research on individual decision-making credibility is less problematic, and the PoW (proof of work) mechanism is introduced in the upper-level planning to suppress the malicious impulse of the node and ensure the credibility of the decision. Under the premise of the degree, the task is evaluated reasonably, and the upper decision model is established with the maximum risk and return amount as the goal. The risk-reward amount generally refers to some extra income that investors get more than the time value of funds due to venture capital investment. This paper thinks that the biggest innovation of block-chain is trust, not money, and it has

decentralized innovation and digitization. Trust provides assurance and support for the reasonable assessment of the tasks in this paper, so the upper objective function is

$$\max P_1 = P_{br} + P_{fl} + P_{te} - P_{com} \tag{16}$$

The specific expressions of each risk return are as follows:

1.  The amount of default risk income generated by the distributed power/power user's predicted deviation from its output/load resulting in poor performance capability.

$$P_{br} = f_{DG} \cdot Y_{DG,user} \cdot \left(\frac{P_T}{P_S}\right) \cdot (1 + \alpha_{DG,user} \cdot P_{\Sigma}) \tag{17}$$

where $f_{DG}$ is a distributed energy derating factor, which is used to describe the influence of external environmental factors on distributed energy output, $Y_{DG,user}$ is the maximum output power of distributed energy or the maximum adjustable output of power users, $P_T$ is the actual output of distributed power or power users, $P_S$ is the predicted output of distributed power or power users, $\alpha_{DG,user}$ is the proportion of distributed energy output, and $P_{\Sigma}$ is the total amount of regulatable resources.

2.  The amount of liquidity risk income generated by poor real-time distributed decision-making in the case of delay, error, and failure in information transmission.

$$P_{fl} = P_{nor} - \frac{\eta \cdot P_{fau}}{\varepsilon} - \left(P_{fau} - \frac{\eta \cdot P_{delay}}{c} - P_{err}\right) \cdot Hash_{(10)} \tag{18}$$

where $\eta$ is the channel capacity, indicating the maximum information rate that the channel can transmit without error, $\varepsilon$ is the repair request, $c$ is the processing request, $P_{nor}$ is the decision benefit under normal communication, and $P_{fau}$ is the decision benefit under fault communication. $P_{delay}$ and $P_{err}$ refer specifically to decision delays under channel delay and channel failure, respectively. $Hash_{(10)}$ is the hash value after hexadecimal conversion to decimal when using SHA256 algorithm. The hash value in the block-chain is a fixed-length, numerically unique, and irreversible message digest that can be used as a unique identifier for the file.

3.  The amount of maturity risk income generated by different blocks in the reverse solution to the SHA-256 problem for competing for the billing rights.

$$P_{te} = P_{DG,0} + \frac{P_{DG,1} - P_{DG,0}}{t_1 - t_0}(t - t_0) \tag{19}$$

where $P_{DG,0}$ is the output power of the distributed power source before competing for the billing right, $P_{DG,1}$ is the output power after the distributed power source competes for the billing right, $t_1 - t_0$ is the calculation time lag, and $t_0$ is the calculated start time.

4.  Compensation for damage caused by user equipment damage and power interruption during the process of correcting the topology matching degree.

$$P_{com} = \beta \cdot \Delta t \cdot \left[(P_{dep,fau} - P_{dep,nor}) + (P_{int,fau} - P_{int,nor})\right] \cdot \left(\frac{e^{r\theta} - 1}{e^{\theta} - 1}\right) \tag{20}$$

where $r$ is the matching deviation, $\theta$ is the individual decision credibility, $\beta$ is the risk value coefficient, and $\Delta t$ is the compensation time difference, which can be expressed as $\Delta t = t_{com,aft} - t_{com,bef}$, $P_{dep,fau}$ is the depreciation expense of the equipment in the fault state, $P_{dep,nor}$ is the depreciation expense of the equipment under normal conditions, $P_{int,fau}$ is the electric energy cost under the interruption of the power supply condition, and $P_{int,nor}$ is the electric energy cost under the normal power supply condition.

The constraints are:

1.  Distributed power supply can regulate output limitation

$$0 \leq P_{DG,user,i} \leq P_{DG,user,\max} - P_N \tag{21}$$

where $P_{DG,user,i}$ is the distributed energy adjustable power of node $i$, $P_{DG,user,\max}$ is the adjustable power upper limit, and $P_N$ is the rated power.

2.  Decision credibility constraint Each block in the block-chain consists of a block header and a block body [30]. The block body is responsible for recording all transaction information for a period of time before, and the block header contains the hash value of the previous block, and the block can be implemented. Most of the functions of the chain, the block header and the block body are connected by the Merkle root. The Merkle root is the root node of the transaction Merkle tree, which can help each node in the block-chain network to quickly verify the existence and correctness of the transaction. Therefore, the introduction of Merkle root in this paper greatly improves the operational efficiency and scalability of intelligent decision-making process while characterizing the credibility of individual decision-making. Merkle is proportional to $P_{cred}$. The larger the value, the higher the credibility of individual decision-making.

3.  Decision credibility $P_{cred}$ calculation formula is

$$P_{cred} = B \cdot Merkle_{(10)} \cdot (x_k w_k + x_e w_e + x_s w_s) \tag{22}$$

where $B$ is the decision evaluation coefficient, and $Merkle_{(10)}$ is the Merkel root value after hexadecimal conversion to decimal using the SHA256 algorithm. It is obtained by hashing the hash of the data block $L_1$ and the hash of the data block $L_2$ and then performing the hash operation. $x_k$, $x_e$, and $x_s$ are the experience, familiarity, and estimation of the scene, respectively. $w_k$, $w_e$, and $w_s$ respectively represent the corresponding proportions.

*4.2. Lower Decision Model*

The problem of multi-task assignment increases exponentially with the increase of the total number of targets, which becomes a multi-parameter and multi-constrained NP-complete problem. Therefore, the influence of user-side distributed new energy on decision-making mode is fully considered in the lower-level control, in order to maximize the degree of user-side control can be used to control the economic and environmental protection of the micro-grid. The lower-level decision-making model is established with the goal of maximum controllable load participation, so as to realize the information interaction and automatic control between the supply and demand sides. Its expression is

$$\max P_2 = (P_{pnt} - P_{ent}) \cdot \omega \tag{23}$$

where $P_{pnt}$ is the controllable load access power value in the nth scene t time period, $P_{ent}$ is the power value that has been accessed in the nth scene $t$ time period, and $\omega$ is the load weight involved in the adjustment.

The constraints are:

1.  Interact power constraints with the distribution network

$$P_{\min-line} \leq P_{DG}(t) \leq P_{\max-line} \tag{24}$$

where $P_{\min-line}$ is the maximum allowed transmission power and $P_{\min-line}$ is the minimum allowed transmission power.

2.  Price cap incentive rules

$$P_{enc} \leq \frac{C_t \cdot (1 + RPI - X)}{1 - r} \cdot Q \tag{25}$$

where $C_t$ is the electricity price cost, $r$ is the profit rate, *RPI* is the electricity price subsidy rate, $X$ is the power regulation efficiency growth rate (day operating cost reduction rate), and $Q$ is the adjustment performance proportional coefficient.

3.   Power supply price constraint on demand side bidding The power supply price of the demand side bid should not exceed its marginal cost. Marginal costs guide an efficient power supply structure and recover all costs.

$$P_{DSB} \leq \frac{\sum\limits_{i=1}^{n} C_{DSB,i} \cdot \Delta P_{DSB,i}}{P_{G,i}} \tag{26}$$

where $C_{DSB,i}$ is the unit compensation price of the demand side bid, $\Delta P_{DSB,i}$ is the demand side bid load reduction, and $P_{G,i}$ is the active input of node $i$.

## 5. Solving Algorithm

### 5.1. Evolutionary Game Algorithm Based on Block-Chain Consensus Mechanism

Evolutionary game theory is based on the bounded rational individual, taking the group as the research object, and believes that the group behavior is realized through dynamic processes such as imitation, learning, and communication between individuals [31]. This theory is a good description of the trends in group behavior and accurately predicts individual group behavior. The evolutionary game theory proposes to make up for the difficulty in solving the complete rationality in the real world and the difficulty in solving the Nash equilibrium in classical game theory. This paper proposes the mechanism of evolutionary game algorithms based on the block-chain consensus mechanism (CM-EGA), and introduces the key technology in the distributed ledger based on the traditional agent evolutionary game algorithm: consensus mechanism, and smoothing imitate the way to update the strategy and solve the upper layer model to improve the real-time performance of the cluster.

Aiming at the problem that the current evolution game stability analysis results are limited to the symmetric payment matrix, this paper introduces the core technology of distributed ledger in the block-chain: consensus mechanism to reduce the information asymmetry, and provide a way to solve the incomplete information evolution game of asymmetric payment matrix.

The distributed ledger in the block-chain can establish a 'trust' ecosystem that satisfies the occurrence and development of economic activities in an asymmetric and uncertain environment. It is not a collusion and complete information disclosure by both parties, but through distributed accounts. The consensus mechanism establishes a trustworthy information interaction platform between the two parties. Each node in the system can see 64-bit hexadecimal information after hash encryption. This paper considers it as a new idea for replicating dynamic equations and evolutionary stability strategies. Enhance the feasibility and credibility of the game results, quantitative analysis of user equilibrium issues, reduce the potential risk of decentralization while reducing the cost of trust.

The update process of this strategy uses a smooth imitation. If and only if the fitness of Agent $i$ is higher, Agent $i$ adopts the strategy $s_j$ of Agent $j$, otherwise the original strategy $s_i$ is adopted. Using $\Omega_i$ to represent the set of neighbors that evolved with Agent $i$, the probability that Agent $i$ mimics the Agent $j$ strategy is

$$\eta_i(s_i' = s_j) = g(f_j - f_i), j \in \Omega_i \tag{27}$$

where, when $x > 0$, $g(x)$ takes a value of 1; when $x \leq 0$, $g(x)$ takes a value of 0.

Since 'proportional imitation' cannot be continuous and balanced, this paper uses the more commonly used improvement method *Fermi − Dirac* function

$$g(x) = \frac{1}{1 + \exp(-x/T)} \tag{28}$$

where $T$ is a constant, representing the rational degree of the player. The probability that the agent $i's$ policy changes from $s$ to $s'$ can be written as

$$\eta_i(s \to s') = \frac{\varphi}{|N_i|} \sum_{j \in N_i(s')} \frac{1}{1 + \exp(-(f_j - f_i)/T)} \tag{29}$$

When $T = 0$, it means complete rational choice, each iteration imitates the most profitable player strategy; when $T > 0$, it means randomly selecting the optimal strategy according to a certain probability; when T tends to infinity, it means completely random selection [32]. In this case, even if Agent $i's$ benefit is higher than Agent $j's$ benefit, Agent $j's$ strategy will still be adopted with a limited probability. This method is called 'smooth imitation'. This paper uses this method to update the strategy in the evolutionary game, which is more in line with the imitation process under actual conditions, and the engineering practical significance is strong.

*5.2. Mixed Game Reinforcement Learning Algorithm*

The lower-level model takes the maximum controllable load participation as the planning goal, fully exploits the potential of the user-side controllable resources, and provides real-time auxiliary services to the system based on optimizing the power configuration, which greatly improves the terminal power efficiency and the system intermittent energy. Access capability and security and stable operation capabilities. The lower layer model is used to quantify the output of the user-side controllable load to eliminate the distributed energy prediction error of wind turbines, photovoltaics, etc., including the distribution network interaction power constraint, price incentive constraint and demand side bidding power supply price constraint, which is a non-convex power system. Supply and demand interaction model, mixed game reinforcement learning algorithms (MGRLA) can solve the non-convex supply and demand interaction model of power system with valve point effect. It is suitable for distributed framework and can effectively solve the demand side under the power market environment. Power system supply and demand interaction model for complex networks.

Based on the multi-agent Stackelberg game, the hybrid game reinforcement learning algorithm uses Q learning as the carrier and the double-loop mobile topology as the framework. The block-chain decentralized block decision and evolutionary game mechanism are introduced to make the interaction between neighbors. The knowledge matrix information interaction formed by the game relationship is fully utilized, and the optimization ability is strong, which can solve the non-convex optimization problem in multi-task decision making with high quality.

5.2.1. Algorithm Description

1. The state value s and the reward value r in the reading environment of each decision subject divided by the double loop topology;

2. Find the de-blocking decision matrix of the block-chain under binary coding

$$O_{iim}^k(s_{iim}^{kj}, a_{iim}) = \partial \cdot \sum_{h \in \Omega_i(k)} (Q_{iim}^k(s_{iim}^{kj}, a_{iim}) - Q_{ihm}^k(s_{ihm}^{kj}, a_{ihm}) \tag{30}$$

where $O_{iim}^k(s_{iim}^{kj}, a_{iim})$ is the m-th binary bit in the i-th knowledge matrix of the *i*-th agent, the action selection made during the *k*-th iteration, $(s_{iim}^{kj}, a_{iim})$ is the state action pair, where the knowledge matrix refers to $Q$ learning store the state action pair's value function $Q(s, a)$ matrix. $\partial$ is the weight value of the block coordination, which is used to reflect different network topologies and properties.

By encoding the action state in two systems, it not only solves the dimensionality disaster problem, but also ensures the calculation accuracy of the continuous optimization problem. The biggest difference between the block decision matrix and the traditional $Q$ learning is that the $Q_{hi}(h \in \Omega_i)$ matrix corresponding to the same node *i* stored in different blocks can be learned interactively, and the

coordinated $Q_{ii}$ matrix corresponding to the node $i$ in the block $\Omega_i$ is obtained, and the block chain is distributed and encrypted. Action selection under the guidance of the ledger.

3. Get the current best strategy

$$\dot{m}_p = \text{argmax}[E_{leader}(\dot{m}), E_{leader}(\dot{m}_{best})] \tag{31}$$

$$s.t. E_{leader}(\dot{m}) = \sum_{i=1}^{n} E_i(\dot{m})$$

$$\dot{m}_i = \underset{m_i \in A_i}{\text{argmax}} E_i(\dot{m}_p, m_i), i = 1, 2, \ldots, n$$

where $p$ is the leader's optional strategy, and $\dot{m}$ is the optimal set of strategies for each follower after the leader adopts strategy $p$. $\dot{m}_i$, $\dot{m}_{best}$, and $\dot{m}_p$ are the optimal strategies for the $i$-th follower, all followers, and leaders. $E_{leader}$ and $E_i$ are the benefit function of the leader, and the $i$-th follower. $A_i$ is the policy domain of the follower.

1. Calculate the reward function

$$R^j_{im}(s^{k,j}_{im}, s^{k+1,j}_{im}, a^{k,j}_{im}) = \frac{c_m}{F^{Best}_i + t_f}, (s^{k,j}_{im}, a^{k,j}_{im}) \in OA^{Best}_i \tag{32}$$

$$F^{Best}_i = \min_{j=1,2,\ldots,j} F^{k,j}_i \tag{33}$$

where $F^{Best}_i$ is the minimum value of the objective function of the optimal individual in the $k$-th iteration of the $i$-th agent, $F^{k,j}_i$ is the j-th individual in the $i$-th agent, the objective function in the $k$-th iteration, and $c_m$ is a positive multiple, $t_f$ is a correction coefficient that guarantees that the reward function is positive, and $OA^{Best}_i$ is the state action pair set of the optimal individual in the $k$-th iteration of the $i$-th agent.

2. Update the knowledge matrix

$$O^{k+1}_{im}(s^{k,j}_{im}, a^{k,j}_{im}) = O^k_{im}(s^{k,j}_{im}, a^{k,j}_{im}) + \alpha \Delta O^k_{im} \tag{34}$$

$$\Delta O^k_{im} = R^j_{im}(s^{k,j}_{im}, s^{k+1,j}_{im}, a^{k,j}_{im}) + \gamma \max_{a_{im} \in A_{im}} O^k_{im}(s^{k+1,j}_{im}, a_{im}) - O^k_{im}(s^{k,j}_{im}, a^{k,j}_{im}) \tag{35}$$

$$m = 1, 2, \ldots, M; j = 1, 2, \ldots, J$$

where $R^j_{im}(s^{k,j}_{im}, s^{k+1,j}_{im}, s^{k,j}_{im} a^{k,j}_{im})$ is the reward function for transitioning from state $s^k$ to $s^{k+1}$ when action $a^k$ occurs, subscript $i$ is the $i$-th agent, and subscript m is the $m$-th binary bit, $\alpha$ is the learning factor, $\gamma$ is the output ratio of the agent, $\Delta O$ is the knowledge increment, and $J$ is the population size in one iteration.

3. Evolutionary game strategy update

$$\eta_i(O^k_{iim}(s^{k,j}_{iim}, a_{iim}) \leftarrow O^k_{ihm}(s^{k,j}_{ihm}, a_{ihm})) = \frac{1}{1 + \exp[(U_i - U_h)/T]} \tag{36}$$

Using the *Fermi − Dirac* rule, the left side of the equation indicates the probability that Agent $i$ will mimic the neighbor Agent $h$ policy [33]. In each iteration, the algorithm looks for Agent $h$ in the block $i$ that has the maximum social benefit of the micro-grid after the controlled load is adjusted. When $U_h > U_i$, the knowledge matrix $O^k_{ihm}$ of the player $i$ in block $h$ is copied with the probability $\eta_i$.

4. Output condition

$$k > k_{\max} \tag{37}$$

where $k_{max}$ is the maximum number of iterations.

5.2.2. Validity Test

Based on the double-loop mobile topology optimization decision-making body, based on the time-of-use electricity price, referring to the electricity price standard of Table 4, the hybrid game reinforcement learning algorithm proposed in this paper is used to quantify the user-side controllable load output as shown in Table 5 below:

**Table 4.** Time-of-use tariff.

| Electricity Price | Time Slot | Buy Electricity Price (yuan/kWh) | Sell Electricity Prices (yuan/kWh) |
|---|---|---|---|
| Valley time | 22:00–5:00 | 0.17 | 0.13 |
| Flat time | 6:00–7:00 11:00–12:00 17:00 | 0.49 | 0.38 |
| Peak time | 8:00–10:00 13:00–16:00 18:00–21:00 | 0.83 | 0.65 |

**Table 5.** Typical moment load side economic ratio.

| Decision Unit | Decision Ratio | | |
|---|---|---|---|
| | 16:00–17:00 | 20:00–21:00 | 23:00–0:00 |
| Washing machine + air conditioner | 14.3% | 18.78% | 29.45% |
| Electric water heater + dishwasher | 9.67% | 19.42% | 14.9% |
| Air conditioning + washing machine + dishwasher | 19.08% | 20.9% | 30.4% |
| Washing machine + microwave + air conditioner + electric water heater | 56.95% | 40.9% | 25.25% |

Tianjin adopts an incentive response mode [34]: pre-cooling of air conditioners from 16:00 to 18:00, preheating of electric water heaters during 18:00–19:00, ensuring comfort while reducing the peak load of the micro-grid from 20:00 to 22:00. During the period from 23:00–0:00, the fan output is large and the electricity price is low. The power of the air conditioner is increased and the power demand of the washing machine and the dishwasher is shifted to the time. The storage capacity of the house is used to store the cooling capacity. At the same time, reduce the load demand of air conditioning during high daytime electricity price period. By translating the load of washing machine and dishwasher, the load peak at night can be reduced. The proportion rule quantified in this paper is consistent with the actual decision-making rule, which can verify the effectiveness of the algorithm.

As shown in Table 6, Tianjin Da-gang Power Plant 4 sets of 32,800 thousand coal-fired thermal power units, from 00:00 on 25 June 2018 to 24:00 on 25 August 2018, 60 days (the peak of summer heavy load), generating 1.513 billion kWh, with an average daily power generation is 25.22 million degrees. Based on the optimization decision-making unit, the proposed algorithm uses the full information interaction of the knowledge matrix to achieve accurate calculation of the decision-making output and efficient configuration of the controllable load, alleviating the complicated information and improving the real-time performance [35–42]. At the same time, the peak power consumption can be reduced by 21.7%, which is 6.1% better than that of multi-agent alone. After optimization, the daily operating cost of the micro-grid can be reduced by 3.54%, avoiding the problem that the original load curve peaks and valleys fluctuate sharply.

**Table 6.** Optimization results under the hybrid game reinforcement learning algorithm.

| Optimization Method | Peak Load Power Consumption (kW·h/day) | Micro-Grid Daily Operating Costs (yuan) |
| --- | --- | --- |
| Multi-agent | 21.28 million | 9635.20 |
| Hybrid game reinforcement learning algorithm | 1974 million | 9294.11 |

### 5.3. Two Level Model Solving Algorithm

In the bi-level programming model, the upper level reassesses the reliability of decision-making individuals, and the lower level quantifies the contribution of controllable load to eliminate prediction errors and calculates the power flow based on the results of the upper level's reliability evaluation and risk-benefit optimization, taking into account relevant constraints, the results are returned to the upper level to solve the risk return of the game, and iterated to convergence. The flow chart of the bi-level programming model is shown in Figure 9.

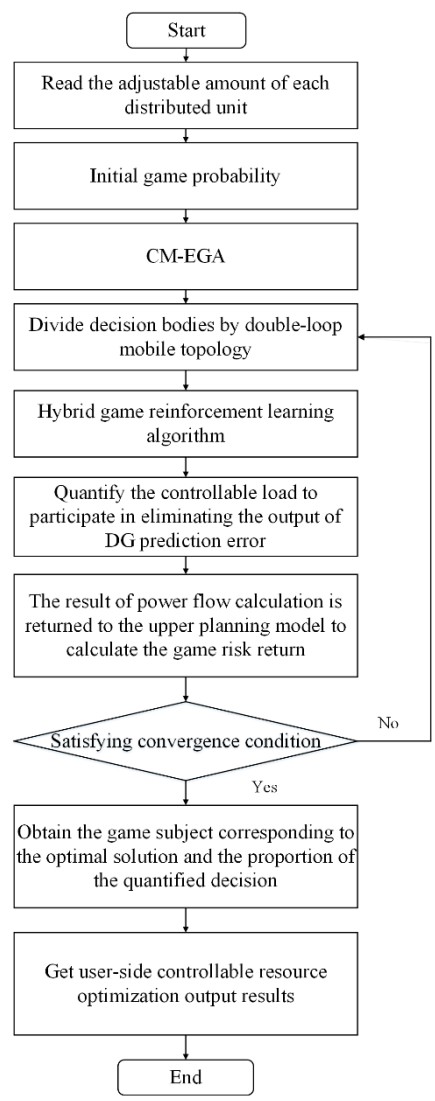

**Figure 9.** Bi-level optimization model flow chart.

## 6. Examples and Analysis of Planning Results

The data source of this paper is Beijing Jin-feng Energy Internet Park. The specific parameters of the device are shown in Table 3.

The regulated loads in the micro-grid include electric water heaters, air conditioners, dishwashers and washing machines, assuming a total of 100 units. The rated power of the electric water heater is 1.45 kW, the upper and lower limits of the hot water temperature are 75 °C and 55 °C respectively; the rated power of the air conditioner is 2.35 kW, and the upper and lower limits of the indoor temperature are 27 °C and 23 °C respectively; the rated power of the dishwasher and the washing machine are respectively 0.5 kW and 1 kW, working hours are 1 h.

The group intelligent decision-making mechanism is adopted to introduce a double-loop mobile topology under MAS to optimize the structure and improve the output power of the distributed decision-making agent in the micro-grid. As shown in Figure 10, due to the power compensation of the inner ring fixed decision maker, there is still a shortage of 0.33 kW. Considering the stability and entropy value, the dishwasher with stable power value and leaving the load margin is involved in the demand side adjustment. Before optimization, the set power of the peak price of electricity at 14:00 is 225.3 kW, and the set power at 17:00 is 273.8 kW. Due to the reasonable evaluation of the load demand of each time period, the consensus mechanism can reduce the set power of 6:00 and 17:00 by 6.5% and 11.2% respectively and improve the economics of micro-grid operation. The lower decision model is guided by price incentive and demand side bidding and uses the hybrid game to strengthen the learning algorithm. The information interaction of the knowledge matrix can reduce the peak and valley fluctuations of the load curve by 5% and improve the stability of the micro-grid operation.

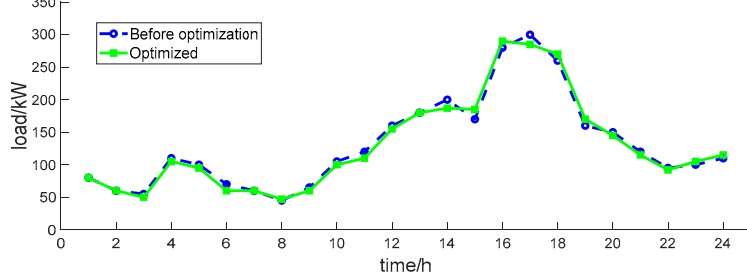

**Figure 10.** Air conditioning + washing machine + dishwasher (corrected BC6) trading chain setting power.

The decision-making bodies in the inner ring are jointly determined by the smart contract and the correction function. Since the probability that the matching degree of the electric water heater and the dishwasher in the transaction chain BC7 is less than 1 exceeds 50%, Therefore, the decision-making body is amended to divide the dishwasher with smaller entropy into another decision-making body, real-time action updates for decision makers. As shown in Figure 11, the underlying hybrid game reinforcement learning algorithm obtains the block-chain block decision matrix interactive learning under binary coding by binary coding the action state, continuously optimizes the action selection in the iterative process, and realizes the accurate calculation of the decision-making output ratio. It can increase the power output of 7.91% at 5:00 in the morning, increase the load adjustment under low electricity price, and advance the preheating time of the electric water heater by 18:00–19:00 1 h in advance, reducing the daily load demand of the micro-grid as a whole. Through group intelligent decision-making, the efficient configuration of controllable load is realized, the utilization of effective information is improved, the problem of complicated information is alleviated, and real-time performance is improved.

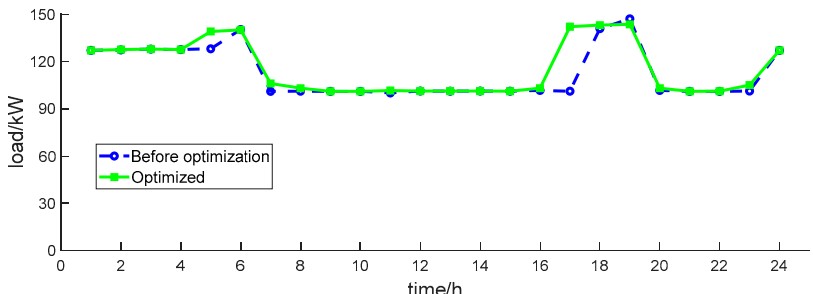

**Figure 11.** Electric water heater (corrected BC7) trading chain setting power.

As shown in Figure 12, after adopting the group intelligent decision-making mechanism, the load of the controllable equipment in each period of the micro-grid changes, and the overall peak-to-valley difference of the load is reduced, and the load curve tends to be smooth. The main reason is that in the mode of block-chain and multi-agent fusion, the design of the double-loop mobile topology is used to correct the decision-making body in real time, and the average number of redundant interactions is reduced by 323 times per day. At 5:00, the decision-maker should be amended as air-conditioning decision-making chain or shift the use time of washing machine and dishwasher, so as to improve the utilization rate of effective information. Using workload consensus mechanism to restrain malicious nodes and ensure the reliability of individual decision-making, the task can be evaluated reasonably, and the real-time decision-making can be improved. Using the lower level hybrid game reinforcement learning algorithm to quantify the load output, and combining the block decision-making under the decentralization of block chain, the accurate calculation of the proportion of decision output and the efficient allocation of distributed energy are realized, and the multi-task execution ability is improved. It shows that the power value increases under the valley price and the load translation or power value decreases under the peak price.

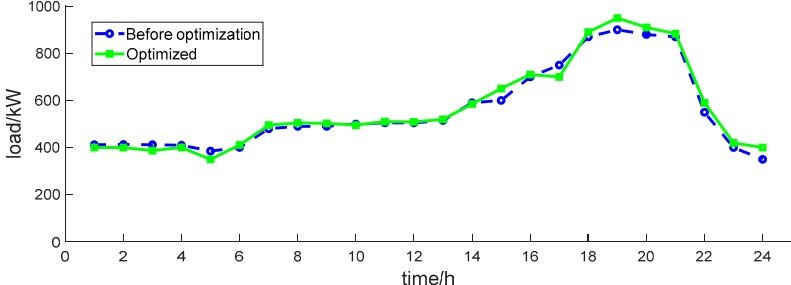

**Figure 12.** Overall load curve for each period.

It can be seen from Figure 13 that after adopting the group intelligent decision-making mechanism, the operating cost of 18:00–21:00 during peak hours can be significantly reduced, and maintain the running cost of the controllable load output in the micro-grid from 22:00 to the next day at 5:00. This is because when the lower level model is solved, the block decision under the decentralization mechanism is introduced into the lower level hybrid game reinforcement learning algorithm to realize the full interaction of the knowledge matrix between the neighbors, and then the load decision ratio of each time period is accurately calculated and fed back to the decision. The correction function performs real-time updates of the decision body selection. That is to say, under the updated decision-making body, the accurate calculation of the proportion of controllable load decisions is made, and the decision-making allocation is optimized to obtain the daily load characteristics with better economic performance.

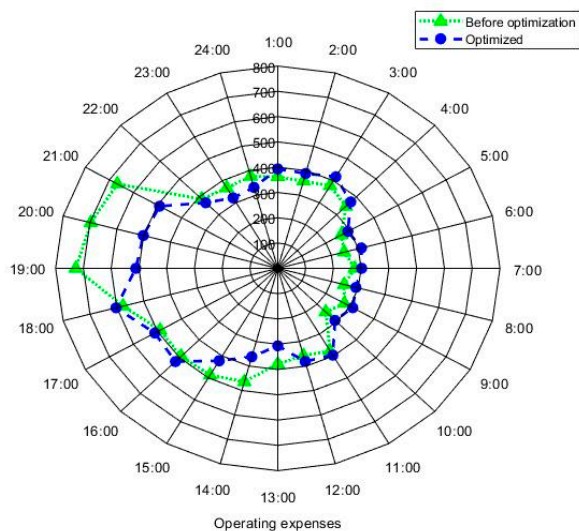

**Figure 13.** Economic benefits of micro-grids at various times.

It can be seen from Table 7 that in the mode of combining block-chain and multi-agent, group intelligent decision-making is used to guide multi-agent to make decision-making subject selection, decision task evaluation, decision task assignment and execution, which can effectively improve the daily operating cost of the micro-grid. In the example of this paper, the use of group intelligence decision-making to perform multi-task can reduce the daily operating cost of the micro-grid by 4.56%, improve the multi-task execution efficiency by 21.56%, and reduce the system energy consumption by 11.06%, Thereby, the rational allocation of the distributed energy in the micro grid is improved, and the output characteristics of the multi-agent are improved.

**Table 7.** Optimization parameters under the group intelligent decision-making mechanism.

| Decision Making Method | Micro-grid Daily Operating Cost (yuan) | Multitasking Execution Efficiency | System Energy Consumption |
| --- | --- | --- | --- |
| Multi-agent | 9846.52 | 62.59% | 41.34% |
| Group intelligence decision | 9397.20 | 84.15% | 30.28% |

## 7. Conclusions

This paper aims to improve the efficiency of supply and demand interactive communication within the micro-grid, optimize decision-making bodies, and reduce energy consumption. Aiming at the three problems of redundant interaction, poor real-time performance and excessive processing information under multi-agent structure, the innovative control strategy of block-chain and multi-agent fusion is proposed. The decentered double-loop mobile topology and the group intelligent decision-making two-layer model can solve the problem in a targeted manner and get the following conclusions:

Firstly, the double-loop mobile topology structure under the decentralized mechanism can reduce the topology complexity, reduce the redundant interaction between agents, and ensure the rationality of the decision-making body. Secondly, through the consensus mechanism to achieve rapid selection of power supply nodes, the communication efficiency can be improved by 8.6% in the decision-making body correction. Based on the improved evolutionary game algorithm of block-chain consensus mechanism theory, through the re-evaluation of decision-making individual credibility and the smooth imitation of bounded rationality, information asymmetry can be effectively reduced, and real-time performance can be improved in decision-making task evaluation. The designed hybrid game reinforcement learning algorithm effectively improves the utilization efficiency of effective information and reduces the amount of information processed by making full use of the information

interaction of the knowledge matrix between neighbors, it can reduce the peak electricity consumption by 21.7%, which is 6.1% more than the multi-agent effect alone. After optimization, the daily operating cost of the micro-grid can be reduced by 3.54%, avoiding the problem that the original load curve peaks and valleys fluctuate sharply. Finally, the topology and two-layer decision model designed in this paper have no significant improvement on the robustness and reliability of the agent. However, it can significantly improve the self-learning ability and communication efficiency of the agent and achieve accurate calculation of the ratio of decision-making output and efficient configuration of distributed energy. At present, this paper only studies the group intelligent decision-making process under the decentralization from the internal of a single micro-grid. In the future, we will further consider the influence of interactive transactions and scheduling between multiple micro-networks on group intelligent decision-making under the mode of block-chain and multi-agent fusion, and explore the feasibility of proposing the novel algorithm proposed in this paper to solve other optimization problems in power system.

**Author Contributions:** All authors contributed to the research in this paper. Z.W. and H.W. conceived and designed the model. Z.S. provided the data. X.F. analyzed the data. W.Y. and P.M. wrote the paper.

**Funding:** This work is supported by the National Natural Science Foundation of China (grant no. 51477099); Key Laboratory of Control of Power Transmission and Conversion (SJTU), Ministry of Education (grant no. 2016AB14).

**Conflicts of Interest:** The authors declare no conflict of interest.

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
