# Peer review of "Research on Micro-Grid Group Intelligent Decision Mechanism under the Mode of Block-Chain and Multi-Agent Fusion"

_energies, doi:10.3390/en12214196_

Round 1

Reviewer 1 Report

Line 44 - Once you have introduced the term and the abbreviation MAS, you should use it in the rest of the paper, otherwise there is no sens in doing it.
After line 48, you should use bullets or numbers (a),(b)... etc for a better understanding of the aspects you are referring to.
Take care about some missing spaces as in Line 51 "operation[2].However", Line 85.
Line 54 - "Design an agent organization that includes both centralized and distributed concepts." You should give some arguments to say why it is required. Are the following lines such an argumentation? it is not clear on a first reading.
Line 63 - "ability of each multi-agent to perform multi-tasking." what you mean "each multi-agent"?, do you have many multi-agent sysems?, it is not what are you talking about in the begining. Again in Line 83.
Line 92. Use consistently the introduction of abbreviation "using P2P peer-to-peer" should be as in Line 44: "using peer-to-peer (P2P)"
Lines 94-99 it is obscure, please re-write to point out the essentials, it is part of your proposal or goals?
Line 101, number 1. The writing fails in accurately underline the contribution, please re-do it taking care of the punctuation.
Line 111, again, the writing is a bit messy, observe the punctuation, commas and dots need to be re-arranged.
Line 138, "works as follows, unlike RPR" unusual construction.
Lines 162-168 The description of the equation's elements need to be reviewed, as r is not defined, and k is defined twice.
Lines 168-173 The description of the equation's elements should be improved. What is iIii?
Lines 171-179 Same as above. You should define every element of the equation, otherwise it makes no sense your descriptions.
Lines 195,196 The flowchart of Figure 2 need to be explained in full, as this is the core of your decision process, you should describe each block in the flowchart to make more readable your proposal. May be insted of having an IF and a WHILE structures you could re-arrange it in just a WHILE loop.
Line 207 The Figure 3 is poorly explained, please add more descriptions related to what it is shown.
Line 235, "analogy consensus mechanism" sometimes you only refer to it as the "consensus mechanism" please choose one an unify the rest, or if they are different, please introduce such a difference.
Lines 248,249. Some terms of the equation are not defined/explained.
Line 251. The first statement looks like incomplete, re-think on what are you trying to state.
Lines 252-258 Some abbreviations used in Figures 4-7 are not introduced.
In general, the explanation of some processes should be extendend, not only equation based but detailed with descriptions. For instance, in Section 2.4 "Validation Check" the decision making process should be extended, indicating the numbers provided by the equations used, as it is well illustrated but not clearly explained accordingly with numbers. It should improve the understanding of your work. One more thing, there is not a clear indication about it is a MAS or a MAS+block chain what it is shown, a good idea should be to illustrate a comparison between both decision-making processes, you do some of it in Table 2, but it is not enough.

Author Response

Response to Reviewer 1 Comments

Point 1: Line 44 - Once you have introduced the term and the abbreviation MAS, you should use it in the rest of the paper, otherwise there is no sens in doing it.

Response 1: The MAS interactive network model and the flexible load model are established in lines 120 to 205. The rest of the control strategy of the article is based on this design.

Point 2: After line 48, you should use bullets or numbers (a),(b)... etc for a better understanding of the aspects you are referring to.

Response 2: Project numbers (a) through (j) have been added to lines 48 to 86, which clearly describe the current state of research and different problems of MAS.

Point 3: Take care about some missing spaces as in Line 51 "operation[2].However", Line 85.

Response 3: I am sorry that I have not noticed the problem with spaces. I have added spaces on lines 51 and 85.

Point 4: Line 54 - "Design an agent organization that includes both centralized and distributed concepts." You should give some arguments to say why it is required. Are the following lines such an argumentation? it is not clear on a first reading.

Response 4: "Design an agent organization that includes both centralized and distributed concepts." This is a research direction of multi-agent systems, not the argument of this paper. The following sentences have pointed out the problems of this control strategy.

Point 5: Line 63 - "ability of each multi-agent to perform multi-tasking." what you mean "each multi-agent"?, do you have many multi-agent sysems?, it is not what are you talking about in the begining. Again in Line 83.

Response 5: "Ability of each multi-agent to perform multi-tasking." that means " the communication efficiency of each multi-agent when making decisions. "Line 83 is the same meaning.

Point 6: Line 92. Use consistently the introduction of abbreviation "using P2P peer-to-peer" should be as in Line 44: "using peer-to-peer (P2P)"

Response 6: Replace 92 lines with "peer-to-peer network".

Point 7: Lines 94-99 it is obscure, please re-write to point out the essentials, it is part of your proposal or goals?

Response 7: Lines 94 to 99 have been rewritten, pointing out the core work of this article.

Point 8: Line 101, number 1. The writing fails in accurately underline the contribution, please re-do it taking care of the punctuation. Line 111, again, the writing is a bit messy, observe the punctuation, commas and dots need to be re-arranged.

Response 8: The contents of lines 101 through 103 and lines 110 through 114 have been rewritten with a focus on punctuation.

Point 9:Line 138, "works as follows, unlike RPR" unusual construction.

Response 9: Removed the sentence of 138 lines: "works as follows, unlike RPR".

Point 10: Lines 162-168 The description of the equation's elements need to be reviewed, as r is not defined, and k is defined twice.

Response 10: Line 248 complements the definition of r.

Point 11: Lines 168-173 The description of the equation's elements should be improved. What is iIii? Lines 171-179 Same as above. You should define every element of the equation, otherwise it makes no sense your descriptions.

Response 11: Lines 260 to 261 complete the expression of the other program and explain . Line 267 explains .

Point 12: Lines 195,196 The flowchart of Figure 2 need to be explained in full, as this is the core of your decision process, you should describe each block in the flowchart to make more readable your proposal. May be insted of having an IF and a WHILE structures you could re-arrange it in just a WHILE loop.

Response 12: From line 277 to line 287 is a complete description of Figure 3, which additionally describes the role of each block in the flow chart.

Point 13: Line 207 The Figure 3 is poorly explained, please add more descriptions related to what it is shown.

Response 13: Lines 290 through 301 are a complete description of Figure 4, supplementing the working principle.

Point 14: Line 235, "analogy consensus mechanism" sometimes you only refer to it as the "consensus mechanism" please choose one an unify the rest, or if they are different, please introduce such a difference.

Response 14: Lines 328 through 330 explain the meaning of " analogy consensus mechanism " and refer to the analog function is the same as the Hash value in the consensus mechanism, and can play a guiding role.

Point 15: Lines 248,249. Some terms of the equation are not defined/explained.

Response 15: Lines 344 through 348 complement the meaning of the variables in the equation, explaining each element.

Point 16: Line 251. The first statement looks like incomplete, re-think on what are you trying to state. Lines 252-258 Some abbreviations used in Figures 4-7 are not introduced.

Response 16: Lines 350 through 361 clearly describe the background and source of the validity check, and the abbreviations in Figures 5 through 8 are also clearly introduced.

Point 17: In general, the explanation of some processes should be extendend, not only equation based but detailed with descriptions. For instance, in Section 2.4 "Validation Check" the decision making process should be extended, indicating the numbers provided by the equations used, as it is well illustrated but not clearly explained accordingly with numbers. It should improve the understanding of your work.

Response 17: Lines 354 and 355 add data source tables, which serve as data support for subsequent experiments.

Point 18: One more thing, there is not a clear indication about it is a MAS or a MAS+block chain what it is shown, a good idea should be to illustrate a comparison between both decision-making processes, you do some of it in Table 2, but it is not enough.

Response 18: Lines 382 through 386 indicate: " All of the models and control strategies described in this paper are under the “MAS+ block-chain” structure " and indicate the limitations of MAS and the advantages of "MAS+ block-chain".

I am very happy with your suggestions on my paper. I am very willing to make changes and improvements. If you have any doubts, please do not hesitate to contact me.

Reviewer 2 Report

Thank you for inviting me as a reviewer for manuscript titled Research on Micro-grid Group Intelligent Decision Mechanism under the Mode of Block-chain and Multi-Agent Fusion. The paper presents a control strategy based on the integration of multi-agent and block-chain in power system.

The paper is really impressive for the efforts made from you to demonstrate the valence of your model. The model is well explained. Methodology is clear. The paper is clearly, concisely, accurately, and logically written. In my opinion the paper is almost ready for the publication in Energies.  But, the paper would be more exiting if you implement below minor improvements:

- Introduction section is well organized.

- Literature review - The research gap and motivation should be clarified. In current version of the paper literature review is presented in a proper way, but authors should add more recent and relevant articles related to problem presented in the paper. The authors need to compare the results in the light of the literature and indicate what new knowledge has been contributed through this research.

- More details about case study should be presented in revised paper.

- Conclusion - Clearly state your unique research contributions in the conclusion section. No bullets should be used in your conclusion section. What are limitations of the proposed model. This should be clarified to the reders. Define more future research directions.

I will review the revised version of the paper with my pleasure. In this stage, the paper is not ready for publication. Before final acceptance of the paper, authors should made above stated issues.

Author Response

Response to Reviewer 2 Comments

Point 1: Introduction section is well organized.

Response 1: I rewrote the introductory section on lines 15 to 30, first presenting the problem to be studied, and then indicating the approach taken and the model built to make it more logical  and organized.

Point 2:Literature review - The research gap and motivation should be clarified. In current version of the paper literature review is presented in a proper way, but authors should add more recent and relevant articles related to problem presented in the paper. The authors need to compare the results in the light of the literature and indicate what new knowledge has been contributed through this research. 

Response 2: 19 to 25 and 36 to 42 of the literature review are new, including the latest articles, which can help readers deepen their understanding of this article and highlight the contribution of this article.

Point 3:More details about case study should be presented in revised paper.

Response 3: Lines 702 to 706 add pre-optimization information, and lines 718 to 724 detail how to binary code the action state.Add data source table.Figure 10 and Figure 11 are the corrections for the decision-making body, Figure 12 is the overall load curve for each period, Figure 13 is the economic benefit of the micro-grid for each period, and the optimization methods used in the four figures have been explained, such as line 733 To line 734, lines 738 through 743, and lines 751 through 755 are shown.

Point 4: Conclusion - Clearly state your unique research contributions in the conclusion section. No bullets should be used in your conclusion section. What are limitations of the proposed model. This should be clarified to the reders. Define more future research directions.

Response 4:Lines 772 through 803 rewrite the conclusion, removing bullet points and enhancing logic with first and second logical connectives, and line 776 indicates the author's outstanding contribution. Lines 793 to 795 illustrate the limitations of the proposed optimization algorithm, and lines 795 to 803 illustrate future research directions.

I am very happy with your suggestions on my paper. I am very willing to make changes and improvements. If you have any doubts, please do not hesitate to contact me.

Round 2

Reviewer 1 Report

You have improved the paper enough to be published, now there are some explanations to make it more readable and understandable. I just insist in the following couple of topics that may be caused a misunderstanding:
- The usage of MAS/multi-agent system, the point is that once you have introduced the term and its abbrevation between parenthesis: (MAS), you should use MAS instead of multi-agent system like in Line 144. May be the unique part where to use the long version is in titles like in Line 206, but in the body of the text you should use MAS.

- In you Response 5, you said: " Ability of each multi-agent to perform multi-tasking." that means " the communication efficiency of each multi-agent when making decisions. "Line 83 is the same meaning."  So, I understand that you are talking about many MAS, not just one, but it it confusing. I think that you are talking about agents belongin to a just one MAS, not many of them. Please make the correction, or explain what are you referring to.

Reviewer 2 Report

I am very happy that the authors have addressed my concerns point by point precisely. No further suggestions come from my side. Therefore, I would like to recommend this manuscript to be published.

Author Response

I thank you very much for your approval of my paper. If you have any questions, please do not hesitate to contact me and thank you again.